# A Gene Signature Identifying CIN3 Regression and Cervical Cancer Survival

**DOI:** 10.3390/cancers13225737

**Published:** 2021-11-16

**Authors:** Mari K. Halle, Ane Cecilie Munk, Birgit Engesæter, Saleha Akbari, Astri Frafjord, Erling A. Hoivik, David Forsse, Kristine E. Fasmer, Kathrine Woie, Ingfrid S. Haldorsen, Bjørn I. Bertelsen, Emiel A. M. Janssen, Einar Gudslaugsson, Camilla Krakstad, Irene T. Øvestad

**Affiliations:** 1Centre for Cancer Biomarkers, Department of Clinical Science, University of Bergen, 5053 Bergen, Norway; erling.hoivik@uib.no (E.A.H.); david.forsse@uib.no (D.F.); camilla.krakstad@uib.no (C.K.); 2Department of Obstetrics and Gynaecology, Haukeland University Hospital, 5053 Bergen, Norway; kathrine.woie@helse-bergen.no; 3Department of Obstetrics and Gynaecology, Sørlandet Hospital Kristiansand, 4604 Kristiansand, Norway; Ane.Cecilie.Munk@sshf.no; 4Section for Cervical Cancer Screening, Cancer Registry of Norway, 0304 Oslo, Norway; bieg@kreftregisteret.no; 5Department of Pathology, Stavanger University Hospital, 4068 Stavanger, Norway; saleha.akbari@sus.no (S.A.); astriifr@gmail.com (A.F.); emilius.adrianus.maria.janssen@sus.no (E.A.M.J.); einar.gudbjorn.gudlaugsson@sus.no (E.G.); Irene.tveiteras.ovestad@sus.no (I.T.Ø.); 6Section for Radiology, Department of Clinical Medicine, University of Bergen, 5021 Bergen, Norway; kristine.eldevik.fasmer@helse-bergen.no (K.E.F.); ingfrid.haldorsen@uib.no (I.S.H.); 7Mohn Medical Imaging and Visualization Centre, Department of Radiology, Haukeland University Hospital, 5021 Bergen, Norway; 8Department of Pathology, Haukeland University Hospital, 5021 Bergen, Norway; bjorn.inge.bertelsen@helse-bergen.no; 9Department of Chemistry, Bioscience and Environmental Technology, University of Stavanger, 4036 Stavanger, Norway

**Keywords:** cervical intraepithelial neoplasia (CIN), cervical cancer, conization, prognostic biomarker, HPV test, gene expression analyses, immune activation, LCK, perforin, CD38

## Abstract

**Simple Summary:**

Through implementation of HPV testing as standard primary screening method, the number of women diagnosed with high-grade cervical intraepithelial neoplasia (CIN2-3) has increased. Although only one third of CIN3 will progress to cancer, conization is standard treatment in high-income countries. The aim of this study was to identify tools with which to predict CIN regression relevant for individualizing treatment within this patient group. We compared the transcriptomic immune-profile from 21 lesions with confirmed regression and 28 lesions with confirmed persistent CIN3. A gene signature with high sensitivity to identify CIN3 lesions that regressed during follow-up was identified. When tested in a cervical cancer cohort (*n* = 239) with available transcriptomic data, a high regression signature score was associated with favorable survival, small tumors, and immune infiltration. This study presents a gene signature with the capacity to predict CIN regression, that may potentially guide treatment, and identifies common disease drivers in CIN and cervical cancer.

**Abstract:**

The purpose of this study was to establish a gene signature that may predict CIN3 regression and that may aid in selecting patients who may safely refrain from conization. Oncomine mRNA data including 398 immune-related genes from 21 lesions with confirmed regression and 28 with persistent CIN3 were compared. L1000 mRNA data from a cervical cancer cohort was available for validation (*n* = 239). Transcriptomic analyses identified *TDO2* (*p* = 0.004), *CCL5* (*p* < 0.001), *CCL3* (*p* = 0.04), *CD38* (*p* = 0.02), and *PRF1* (*p* = 0.005) as upregulated, and *LCK* downregulated (*p* = 0.01) in CIN3 regression as compared to persistent CIN3 lesions. From these, a gene signature predicting CIN3 regression with a sensitivity of 91% (AUC = 0.85) was established. Transcriptomic analyses revealed proliferation as significantly linked to persistent CIN3. Within the cancer cohort, high regression signature score associated with immune activation by Gene Set enrichment Analyses (GSEA) and immune cell infiltration by histopathological evaluation (*p* < 0.001). Low signature score was associated with poor survival (*p* = 0.007) and large tumors (*p* = 0.01). In conclusion, the proposed six-gene signature predicts CIN regression and favorable cervical cancer prognosis and points to common drivers in precursors and cervical cancer lesions.

## 1. Introduction

Almost all cervical intraepithelial neoplasia (CIN) and cervical cancers are caused by infection with high-risk human papilloma virus (hrHPV) [1]. Most HPV infections are transient and cleared within a year [2]. From initial HPV infection, it takes between 10–30 years to develop invasive cancer [3,4,5]. This slow progression, through several levels of cellular changes, makes cervical cancer a disease that could be marginalized with optimized screening. CIN is considered a premalignant disease, even though only 1% of low-grade dysplasia (CIN1) [6] and 30% of high-grade dysplasia (CIN2-3) will progress to cancer [4]. Approximately, 20–40% of CIN2-3 will regress [6,7,8,9]. Despite this knowledge, all women with CIN2-3 are encouraged to undergo surgical treatment, mainly by removing a cone-shaped piece from the cervix, with risk of complications such as bleeding, infection, perforation, and cervical stenosis leading to dysmenorrhea, risk of occult endometrial cancer, and difficulties in follow-up regarding adequate cytology by positive HPV test [10,11]. The most severe late complication, however, is the increased risk of preterm birth, potentially leading to significantly higher neonatal morbidity and mortality, in addition to a higher number of adverse obstetric sequelae [12,13,14]. Consequently, a significant number of women, who would never be diagnosed with cervical cancer, end up being treated as high-risk CIN leading to psychological distress and potential overtreatment.

Several countries are currently in the process of substituting primary cytology screening with a primary hrHPV-test [15]. Extensive scientific studies show that HPV-based screening yields 23–27% higher sensitivity for detecting high-grade precancerous lesions compared to cytology, consequently preventing more cervical cancer [16,17]. However, more sensitive screening methods also result in higher detection rates of CIN2-3 lesions with a low risk of cancer development [18]. In addition, changed sexual behavior during the last decades, with younger age at first sexual intercourse and higher number of sexual partners, leads to higher exposure to sexually transmitted infections including HPV [19,20,21]. Furthermore, a steadily increasing HPV vaccinated population reaching screening age–presumably with less aggressive HPV infections and lower disease prevalence, is already causing challenges for clinicians in lack of adequate algorithms. Facing these aspects, robust prognostic biomarkers are urgently needed both for better risk stratification and more personalized screening approaches.

Due to poor reproducibility of CIN2, the College of American Pathologists and American Society for Colposcopy and Cervical Pathology strongly recommend implementation of Lower Anogenital Squamous Terminology (LAST), replacing CIN1, 2, and 3 with the categorization of low-grade and high-grade squamous intraepithelial neoplasia (LSIL and HSIL) [22,23]. Dealing with only two categories, the observational procedure will be more clinically challenging, as there will only be one step between low-grade CIN and cancer. Thus, prognostic biomarkers and adequate algorithms will be beneficial for the gynecologists, currently having frequent follow-ups in women with harmless HPV infections and low risk CIN lesions, as well as for the pathologists, currently experiencing a large proportion of normal/low grade biopsies or CIN2-3 with high probability of regression. A better risk stratification will also impact the society, reducing unnecessary health care spending both for follow-up and treatment of low-risk lesions, and reducing the burden of potential complications by over-treatment.

Despite steadily improving screening strategies, the prevalence of cervical cancer has remained stable over the last decades (Cancer Research UK, https://www.cancerresearchuk.org/, accessed on 6 September 2021; Cancer Registry of Norway, https://www.kreftregisteret.no/, accessed on 6 September 2021). If detected at a late stage, effective treatment options are limited, and prognosis is poor for cervical cancer patients [24]. HPV infection is both the cause and the main driver in cervical carcinogenesis in precancerous as well as cancer lesions. Research on common prognostic factors for regression/progression of precancerous lesions and disease outcome in cancer could thus yield new and valuable information. Immune and stromal cells dominate the microenvironment in CIN and cervical cancer. In the last decade, a better understanding of HPV tumor–host immune system interactions (reviewed in [25]) and the development of new immune checkpoint targeting strategies have led to renewed interest in immunotherapy for cervical cancer patients. To aid in the growing challenge of reducing overtreatment of CIN2-3 lesions (induced by, e.g., implementation of HPV-based screening and a higher proportion of vaccinated women), we aimed to identify immune-related biomarkers that predict CIN regression. By exploring the identified prognosticator of persistent disease in a large cancer cohort, we aimed to identify common carcinogenic drivers in precursor and cancer lesions and subsequently detect reliable biomarkers that can identify CIN3 lesions likely to progress into cancer.

## 2. Materials and Methods

### 2.1. Patient Cohorts and Biospecimen Collection

The CIN population consisting of 657 women, was included prospectively at the gynecologic outpatient clinic, Stavanger University Hospital (SUS), between 2007–2009 (cohort 1) and 2015–18 (cohort 2) as previously described [7,26,27]. The study was approved by the Regional ethical committee (REK number 2016/805, 2019/264, 2012/1292 and 2020/10399), and written informed consent was obtained by inclusion. Stavanger University Hospital is the only referral hospital for CIN treatment for a population of 370,000. The patients were all referred for further diagnostic evaluation of atypical cervical cytology and/or persistent HPV, according to the former national guidelines from the Norwegian cervical cancer screening program. In total, 333 women diagnosed with CIN2-3 in cervical biopsies were treated by cone excision, median 101 (23–1212) days after the diagnostic biopsy. In total, 63/333 (19%) of the women with proven CIN2-3 in punch biopsies, were diagnosed with only CIN1 or normal tissue in the cone biopsy and defined as regression cases. Two expert pathologists in the field of gynecology, blinded for each other’s diagnoses, evaluated all hematoxylin and eosin (HE) stained slides supported by Ki-67 and p16 immunohistochemical (IHC) staining. Forty-nine formalin fixed paraffin embedded (FFPE) biopsies evaluated preoperatively to represent CIN3 were used in the current study. Persistent CIN3 was defined by diagnosis of CIN3 both in the diagnostic and in the cone biopsy. Following these criteria, 21 were regression cases and 28 persistent CIN3. All patients were extensively followed up after cone excision. Median follow-up time after cone excision was 1886 days (range 119–5173 days) (Appendix A). No patients with confirmed regression in the cone excision specimen experienced recurrence with high-grade CIN (CIN2/3) during follow-up.

### 2.2. RNA/DNA Extraction and p16 Immunohistochemistry

FFPE tissue was used for isolation of RNA and DNA using the miRNeasy FFPE kit (Qiagen) and the E.Z.N.A® Tissue DNA Kit (Omega Bio-tek Inc., Norcross, GA, USA) for cohort 1 and “Recover all total nucleic acid isolation” kit (Thermo Fisher Scientific, Waltham, MA, USA) for cohort 2. For all three methods, isolation was performed following the manufacturer’s instructions. RNA was isolated from 5 µm thick sections comprising the most severe dysplastic area of the epithelium and the adjacent stroma, marked by an expert pathologist, supported by p16 and Ki67 staining. To ensure continuous presence of CIN3, areas adjacent to the sections used for DNA/RNA isolation were HE-stained and evaluated by the expert pathologist.

### 2.3. Functional RNA Quantification and RNA Reverse Transcription

To quantify the exact amplifiable RNA concentrations from the FFPE samples, a one-step RT-qPCR procedure was applied (LightCycler^®^ 480 System, Roche Diagnostics, Rotkreuz, Switzerland) to measure the RNA concentration with TaqMan Fast Advanced Master mix together with a TaqMan probe specific for the housekeeping gene *GUSB* (both ThermoFisher Scientific, Waltham, MA USA). To generate target gene standard curves, fourfold dilution series (range of 0.05–50 ng/µL) of a commercially available standard (HL-60, 100 ng/µL total RNA) was used. The RNA concentration for each lesion was calculated, by comparing the mean Ct of triplicates measured for test samples to the Ct measured for the different HL-60 dilutions of the standard curve. The Vilo Mastermix cDNA synthesis kit (Thermo Fisher Scientific) was used for transcription of 10 ng total RNA, as calculated by functional RNA quantification. p16 IHC scoring was performed by using a scale ranging from 0.0 (negative) to 1.0 (strong positive) depending on the proportion of positivity in the epithelium (exclusive basal cells). P16 scoring was available for cohort 1 (*n* = 32). The scoring was dichotomized; staining index 0.0–0.9 was defined as weak to moderate, and 1.0 as strong positive (strong expression through all epithelial layers). The extent and degree of immunopositivity was scored blinded and independently by two pathologists. In case of discrepancy >10%, which occurred in 8% of all cases, a consensus recount was performed.

### 2.4. HPV Testing

Biopsies from cohort 1 were genotyped by using Linear Array (Roche Molecular Systems, Roche Diagnostics, Mannheim, Germany), while biopsies from cohort 2 were genotyped using the InnoLipa HPV detection system (Fujirebio Europe N.V., Gent, Belgium). Both are line probe assays based on reverse hybridization principle for detection of both high- and low-risk, in addition to possible high-risk HPV types.

### 2.5. Next Generation Sequencing (NGS)

The Oncomine™ Immune Response Research Assay targets and quantifies the expression levels of a panel of 398 immune response related genes, including 10 housekeeping genes, using Next Generation Sequencing (NGS), Ion Torrent. Automated library preparation of 2 × 32 samples, each containing 10 ng/µL cDNA, were performed in batches of four libraries of eight samples, according to the Ion Chef system protocol. The library concentrations were measured by the Ion Library TQMN quantitation kit. The Ion OneTouch™ 2 System was used to prepare the enriched, template-positive Ion PI™ Ion Sphere™ Particles (ISPs). The samples were diluted to 100 pM prior to PCR on the Ion OneTouch™ 2 Instrument, using the Ion PI™ Hi-Q ™ OT2 200 kit. A quality control of template-positive ISPs was performed on a Qubit™ 2.0 Fluorometer by using the Ion Sphere™ Quality Control Kit. Enrichment of ISPs was performed by using the Ion OneTouch™ ES instrument. Target sequencing was performed on an Ion Proton instrument by using the Ion PI™ Hi-Q ™ Sequencing 200 chemistry and an Ion PI ™ chip. Subsequently, the sequencing results were downloaded to the Affymetrix Transcriptome Analysis Console (TAC) for further analyses. Mean housekeeping scaled log2 count data from the 398 genes in the Oncomine Immune Response panel, were obtained from the Torrent Suite™ Software. All kits, reagents and software were provided by ThermoFisher Scientific (Waltham, MA, USA) if not otherwise specified.

### 2.6. Gene Expression Analyses

Differentially expressed genes were called by the Feature Subset Selection (FSS) method within the JExpress software (www.molmine.com, accessed on 21 October 2020) [28]. The FSS ranking method was set to individual ranking to score the genes independently based on how they separated between groups (e.g., CIN3 persistence versus regression). Gene set enrichment analyses (GSEA) were performed within the JExpress software comparing lesions with persistence versus regression of CIN3. Scoring method for GSEA was Golub (signal-to-noise), and permutations were performed on genes. C2 (curated gene sets), C5 (gene ontology gene sets), and Hallmarks gene set collections of the Molecular Signature database v4.0 (MSigDB, Broad Institute, Cambridge, MA, USA) were queried for enriched gene sets [29].

### 2.7. Creating a Six-Gene Signature

FSS analyses were performed to identify differentially expressed genes (DEGs) between CIN3 regression (*n* = 21) and persistent CIN3 (*n* = 28) lesions. Six genes matched the criteria of *p* < 0.05 and fold change <−1.75 or >1.75, consisting of five upregulated genes and one downregulated gene in CIN3 regression lesions. A signature score for each patient was created by subtracting the expression value of the downregulated gene from the sum of expression values of the five upregulated genes. A high signature score should thus associate with CIN3 regression.

### 2.8. Cancer Cohort

#### 2.8.1. Patient Characteristics and Biospecimen Collection

The cancer cohort study was approved by the Regional ethical committee (REK-number 2018/591 and 2014/1907) with written informed consent from all patients. All patients were diagnosed and treated at the Haukeland University Hospital (HUS). HUS is a referral hospital for patients in Hordaland County in Western Norway, representing approximately 10% of the Norwegian population with similar patterns of incidence and prognosis as from whole of Norway (Cancer Registry of Norway, http://kreftregisteret.no, accessed on 6 September 2021). All patients admitted to HUS during 2001 until 2017 with available fresh frozen tumor tissue eligible for mRNA profiling were included in this study. Extensive clinical- and histopathological data from primary diagnosis and follow-up were assessed for all recruited patients. All patients were clinically staged following the International Federation of Gynecology and Obstetrics (FIGO) 2009 criteria. Formalin fix paraffin embedded (FFPE) tissue with corresponding HE-stained sections were collected from hospital archives for histopathological evaluation and IHC as previously described [30]. For IHC, the FFPE tissue was mounted in tissue microarrays as previously described [31]. Histological type and grade, depth of invasion, inflammatory reaction, and vascular space invasion were assessed by an expert pathologist, as previously described [31]. MRI of the pelvis was performed on 152 of the patients at primary diagnostic work-up and included T2-weighted sequences acquired in two orthogonal planes. These were used to measure maximum tumor diameter on the slice depicting the largest maximum tumor diameter, as previously described [30]. Disease-specific survival (DSS) was defined as time from primary treatment until death caused by cervical cancer or end of follow-up. An expert pathologist evaluated tumor cellularity on HE-stained sections from the fresh frozen tumor biopsies. Biopsies were included if tumor content was more than 50% and preferably 80%. Total RNA was extracted from the fresh frozen tissue using the All-Prep DNA/RNA Mini Kit (Qiagen, Hilden, Germany) according to manufacturer’s instructions. Quality and yield of total RNA was assessed by spectrophotometry (Nanodrop 1000, ThermoFisher Scientific, Waltham, MA, USA) and Bioanalyzer 2100 (Agilent, Santa Clara, CA, USA).

#### 2.8.2. Gene Expression Profiling

mRNA expression profiles were generated by the L1000 approach [32] for 239 patients. The L1000 expression data were obtained through an algorithm that extrapolates the expression of 978 directly measured (landmark) genes by a method involving ligation-mediated amplification and fluorescent labelling to generate a transcriptional profile of 12,328 genes in the full L1000 dataset [32]. Replicate-collapsed z-scores (level-5 data) were applied for subsequent L1000 analysis. Of the six genes within the identified gene signature in the precursor lesions, all genes except *CCL3* overlapped with the L1000 data creating a 5-gene signature within the cervical cancer cohort. Following the same procedure as for the CIN cohort, a signature score was created for each cervical cancer patient. Optimal gene-signature cut-off values for dichotomization used in GSEA and Kaplan–Meier survival analyses were identified from ROC curves by applying the Youden index [33]. GSEA analyses were performed as described for the CIN cohort. An immune infiltration score was calculated for each patient with available L1000 data by using R version 3.6.3 (Massachusetts, USA) with the ESTIMATE (Estimation of Stromal and Immune cells in MAlignant tumor tissue using Expression) package version 1.0.13 [34].

### 2.9. Statistical Analyses

Statistical data analyses were performed within the Software package SPSS Statistics (Statistical Package of Social Science) version 27.0 (IBM, Armonk, NY, USA). All probability values were two-sided and considered statistically significant if <0.05. Multiple testing correction was calculated by the Benjamini–Hochberg method. For categorical variables, correlation between groups was assessed using Pearson χ2 or Fisher’s exact test as appropriate. For continuous variables, the Mann–Whitney U or the Kruskal–Wallis test was applied as appropriate. Spearman correlation was applied for detection of non-parametric relationships between pairs of continuous variables. Patient survival analyses were performed by using the Kaplan–Meier (product-limit) method, and survival differences were calculated by the log-rank test (Mantel–Cox). Receiver operating characteristics (ROC) analyses were utilized on the gene-signatures to compare performance related risk groups. Optimal gene-signature cut-off values for prediction of CIN3 regression and cervical cancer survival were identified from ROC curves by applying the Youden index [33] with regression as outcome in the CIN cohort and disease-free survival as outcome in the cancer cohort.

**Table 1 cancers-13-05737-t001:** Distribution of clinicopathological characteristics for all CIN patients included in this study. The number of cases in each group is given followed by percentage for each row in parenthesis.

	Cone Excision Diagnosis		
	CIN3 Regression	Persistent CIN3	*p*-Value
	*n* = 21	*n* = 28	
Last cytology before biopsy			0.71 ^a^
AGUS	0 (0)	1 (100)	
ASC-H	4 (36)	7 (64)	
ASCUS	0 (0)	1 (100)	
HSIL	10 (42)	14 (58)	
LSIL	6 (60)	4 (40)	
Normal	1 (50)	1 (50)	
HPV Type in Biopsy			0.79 ^a^
HPV 16	9 (39)	14 (61)	
HPV18	2 (40)	3 (60)	
HPV 31	1 (50)	1 (50)	
HPV 33	4 (36)	7 (64)	
HPV 35	2 (100)	0 (0)	
HPV 39	1 (50)	1 (50)	
HPV 52	2 (50)	2 (50)	
Age at diagnosis			0.19 ^b^
≤29	8 (33)	16 (67)	
>29	13 (52)	12 (48)	
Interval between cytology and biopsy			0.32 ^b^
≤41	12 (50)	12 (50)	
>41	9 (36)	16 (64)	

^a^ Pearson’s *χ*^2^ test. ^b^ Mann–Whitney U-test.

**Figure 1 cancers-13-05737-f001:**
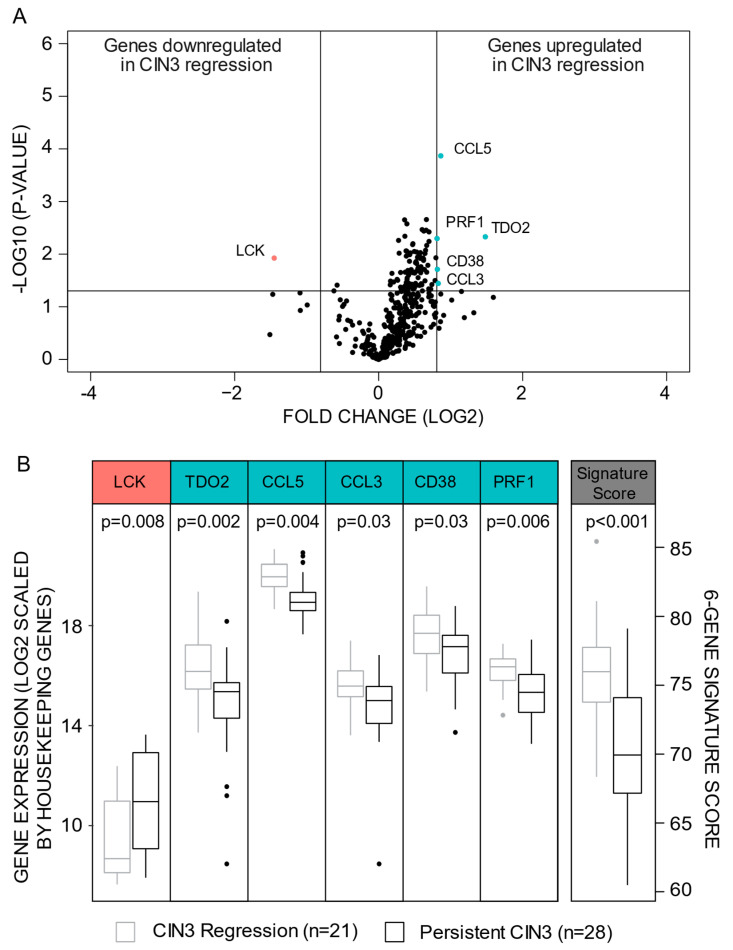
Identification of a CIN regression signature. (**A**) Distribution of differentially expressed genes as defined by the criteria of *p* < 0.05 and fold change <−1.75 or >1.75. (**B**) Distribution of log 2 expression levels (scaled by housekeeping genes) of the six signature genes and the signature score in lesions of confirmed CIN3 regression versus persistent CIN3. The Man–Whitney U test was applied when if the distribution of the genes were different in CIN3 Regression versus Persistent CIN3. Abbreviations: CIN: Cervical intraepithelial Neoplasia.

## 3. Results

### 3.1. A Six-Gene Signature Predicting CIN3 Regression

No statistical differences in cytology prior to biopsy, HPV type, age, interval between last cytology and biopsy as well as biopsy cone interval between patients with CIN3 persistence or regression were detected (Table 1). By FSS, six genes matched the criteria of being significantly differentially expressed between CIN3 persistent versus CIN3 regression lesions (Figure 1A, Appendix A). *TDO2* (*p* = 0.005), *CCL5* (*p* < 0.001), *CCL3* (*p* = 0.04), *CD38* (*p* = 0.02), and *PRF1* (*p* = 0.005) were upregulated, whilst *LCK* (*p* = 0.008) was downregulated in CIN3 regression lesions (Figure 1B, Appendix A).

A signature score was calculated for each patient (Figure 2A, for details: See Methods section). Although none of the genes remained significantly differentially expressed after multiple testing (FDR < 0.05), they displayed strong predictive power when considered together as a signature. With an optimal cut-off of 73.1408, 22 patients were classified with low and 27 with high signature score yielding an area under the ROC curve (AUC) of 0.85, a sensitivity of 91% and a specificity of 74% for predicting CIN3 regression (Figure 2B). A strong negative correlation between the six-gene regression signature and p16 protein expression was detected (Figure 2C).

### 3.2. Persistent CIN3 Associates to Proliferation

In GSEA analyses, within the Hallmark gene sets, “E2F targets genes” and “G2M checkpoint genes” were significantly enriched in persistent CIN3 (Appendix A). Within the C2 curated gene sets, nine out of the top twenty gene sets were related to aggressive cancer or cancer proliferation (Appendix A). Gene sets related to microtubules were significantly enriched in persistent CIN3 within the GO collection (Appendix A).

**Figure 2 cancers-13-05737-f002:**
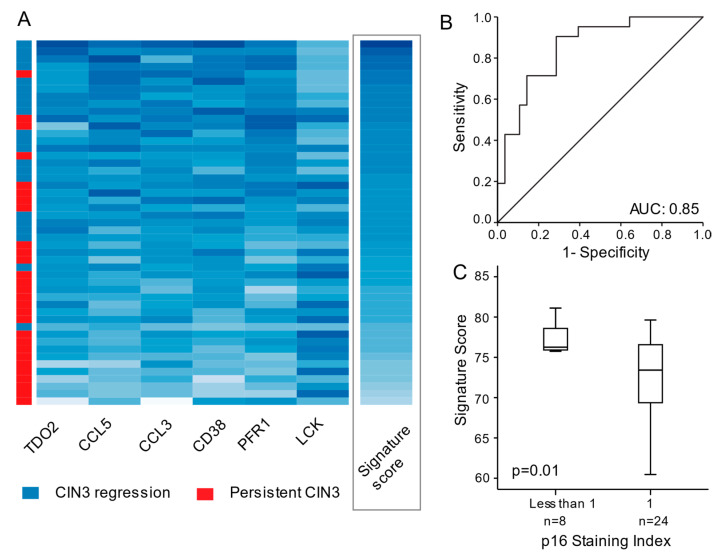
(**A**) Heatmap illustrating signature score for each gene (column) within each individual patient (row) with risk group defined by red or blue color as indicated. Increasing darker blue color represents increasing gene expression level. In the last column, increasing darker blue color illustrates increasing signature score for each patient sorted by highest to lowest signature score. (**B**) Receiver operating (ROC) curve reflecting the sensitivity, specificity, and area under the curve (AUC) for the gene signature to predict risk group within the 49 precursor lesions. (**C**) Immunohistochemistry p16 Staining Index compared to the signature score.

### 3.3. High Regression Signature Score Associates to Favorable Survival and Less Aggressive Features within Cervical Cancer Patients

Next, we sought to characterize the prognostic value of the identified six-gene signature within a cervical cancer cohort with available transcriptional, clinicopathological and follow-up data on 239 patients (Table 2). Within the cancer cohort validation gene set, expression data for *CCL3*, was not available. Consequently, a five-gene signature was created for each cervical cancer patient (For details: See Methods). The five-gene signature strongly correlated to the six-gene signature (Appendix A, Pearson correlation coefficient of 0.98, *p* < 0.001), with an AUC of 0.86 (*p* < 0.01) and the same sensitivity and specificity to predict CIN3 regression as the six-gene signature (Appendix A). We thus concluded that further analyses on this five-gene signature within the cancer cohort was justified.

High signature score, which associated to CIN3 regression in the CIN cohort, correlated to favorable survival (Figure 3A, *p* = 0.007), MRI derived maximum tumor diameter of less than 2 mm (Figure 3B, *p* = 0.01), depth of invasion 7 mm or less (Figure 3C, *p* = 0.03), and no vascular space invasion (Figure 3D, *p* = 0.04) in the cancer cohort.

**Table 2 cancers-13-05737-t002:** Distribution of available clinicopathological characteristics for the cervical cancer patients according to regression signature score. The number of cases in each group is given followed by percentage for each row in parenthesis.

Variables (*n*) ^a^	Regression Signature Score		*p*-Value ^b^
	Low Score (*n* = 91)	High Score (*n* = 148)	
Median age (*n* = 239)			0.98
<44 years	42 (38)	68 (62)	
≥44 years	49 (38)	80 (62)	
BMI (*n* = 236)			0.17
<25	42 (34)	83 (66)	
≥25	47 (42)	64 (58)	
FIGO-09 stage (*n* = 239)			0.27
I-IB1	40 (34)	76 (66)	
IB2-IV	51 (42)	72 (58)	
Histologic type (*n* = 239)			0.15
Squamous cell carcinoma	59 (35)	112 (65)	
Adenocarcinoma	24 (50)	24 (50)	
Other histologic type	8 (40)	12 (60)	
Histologic grade (*n* = 225)			0.06
Grade 1/2	58 (34)	111 (66)	
Grade 3	27 (48)	29 (52)	

Statistically significant *p*-values (*p* < 0.05) are in bold. ^a^ number of cases with available data for each variable. ^b^ Pearson’s *χ*^2^ test.

### 3.4. High Regression Signature Score Is Significantly Associated with Immune Activation in Cervical Carcinomas

To characterize activated pathways according to signature score, tumors with high versus low signature score were compared by GSEA. Within the Hallmarks, C2 and C5 gene set collections, tumors with high signature score were enriched for gene sets associated with immune activation and response to inflammation (Appendix A, respectively). Correspondingly, by histopathological inspection, tumors with high signature score associated with elevated inflammatory reaction (*p* < 0.001, Figure 4A) and increased expression of the major histocompatibility complex (MHC) Class II receptor, HLA-DQB1 [30] (*p* = 0.004, Figure 4B). To further investigate the association between signature score and immune activation, we calculated an Immune Score based on the ESTIMATE algorithm (For details: See Methods). Increasing signature score was significantly associated with increasing Immune Score (Shearman correlation = 0.69, *p* < 0.001, Figure 4C).

**Figure 3 cancers-13-05737-f003:**
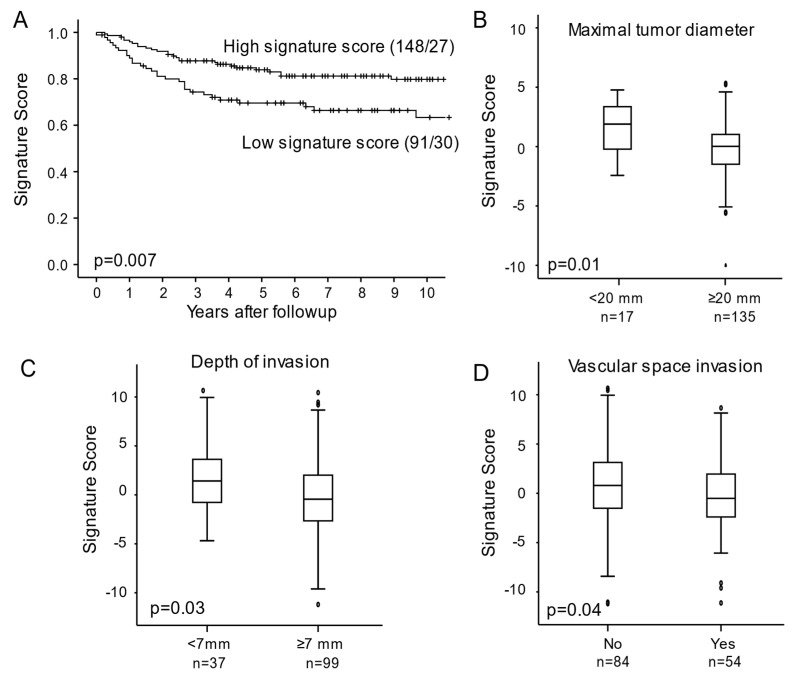
(**A**) Disease specific survival for cervical cancer patients relative to signature score represented by Kaplan Meier curve with probability values for Mantel Cox log-rank test that compares categories. The number of patients and events are given within parentheses (patients/events). Signature score in relation to MRI derived maximal tumors diameter (**B**) histopathological depth of invasion (**C**) and, vascular space invasion (**D**) in tumors with available data.

## 4. Discussion

Despite solid evidence that around 50% CIN2 [35] and 30% of CIN3 lesions [6] will regress spontaneously over time, no systematic test for identifying these cases exists in current clinical practice. Consequently, women with low risk of cancer are offered the same follow-up and treatment regime as those with high risk. Although punch biopsies are included in the diagnostic work-up, so far, its potential to reveal features predicting risk-groups by transcriptomic profiling have not been exploited. To investigate this potential, we extracted mRNA from 21 CIN3 lesions with confirmed regression and 28 CIN3 lesions with confirmed persistence. By differential gene expression analyses, we identified a six-gene signature that can predict CIN3 regression with high accuracy. To our knowledge, this is the first study proposing a transcriptomic-based prognostic test, that may serve as a tool for stratifying CIN3 patients into treatment groups without the need for additional sampling.

**Figure 4 cancers-13-05737-f004:**
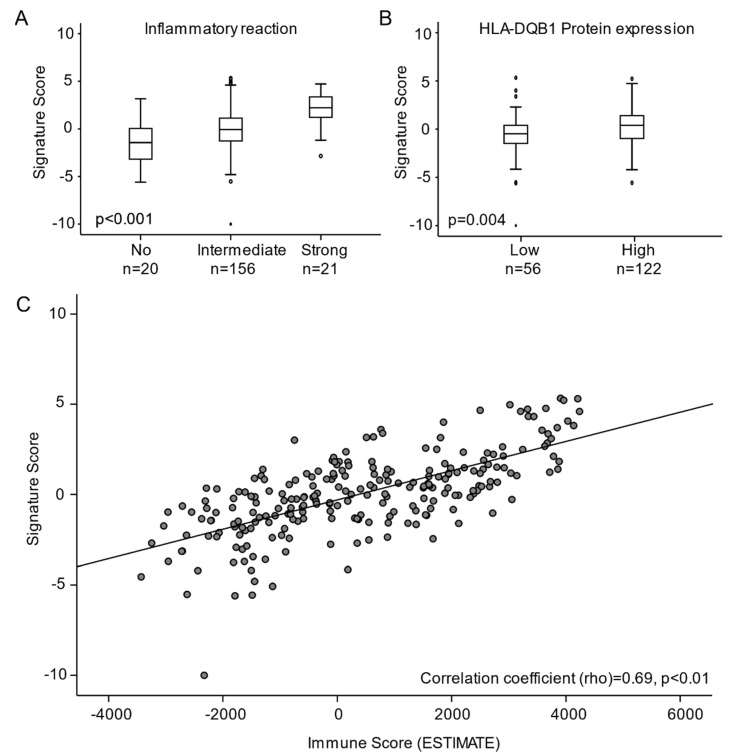
Signature score in relation to (**A**) inflammatory reaction (**B**) HLA-DQB1 protein expression and (**C**) immune score (ESTIMATE). Abbreviations: ESTIMATE: Estimation of Stromal and Immune cells in MAlignant tumor tissue using Expression.

The biological mechanisms determining whether the CIN3 lesions will persist or regress may prove relevant for understanding carcinogenic processes — also in the cervical cancer setting. Genes that are highly expressed in persistent CIN3 could be early drivers of cervical cancer and could serve as prognostic markers and/or therapeutic targets in cervical cancer. Within the six-gene signature, *LCK* was the only gene upregulated in persistent CIN3. Lymphocyte-specific protein tyrosine kinase (LCK) is a member of the Src family of tyrosine kinases predominantly expressed on T cells to regulate T cell development and homeostasis [36]. However, LCK has also been found expressed in several solid tumors including brain [37], breast [38], prostate [39], and colorectal [40] cancer. In breast cancer, LCK promotes angiogenesis and tumor progression [41], indicating an oncogenic role in line with our findings. So far, LCK expression has not been described in cervical precursor nor cancer lesions. Thus, a thorough characterization of its potential as prognostic and/or predictive marker in the precancer and cancer setting is warranted.

Genes that are highly expressed in CIN3 regression as compared to persistent CIN3 could be potential prognostic markers for CIN3 regression and of less aggressive cancer. *TDO2*, *CD38, CCL5*, *CCL3*, and *PFR1* were upregulated in CIN3 regression lesions. Tryptophan 2,3-dioxygenase (TDO) is a heme enzyme encoded by the *TDO2* gene that catalyzes the first and rate-limiting step of tryptophan degradation along the kynurenine pathway [42]. Tryptophan depletion has been linked to aggressive cervical cancer [43] and impaired immune responses by restraining T cell proliferation and restricting tumors immune cell infiltration [44]. Within the TCGA cohort, *TDO2* expression associates with poor survival (*p* = 0.003, https://www.proteinatlas.org/, accessed on 11 August 2021). Altogether, the above reported studies indicate TDO as a tumorigenic driver. Yet our results revealed *TDO2* as significantly higher expressed in the CIN3 regression lesions. In a recent study, TDO expression was detected in infiltrating leukocytes in the adjacent stroma of CIN and invasive cervical cancer [45]. Leukocytes promote a pro-inflammatory environment, which is crucial in triggering the adaptive immune response on HPV infected cells [46]. Thus, in cervical precursor and cancer lesions, TDO expression may primarily be a marker for activated adaptive immune responses, which could in turn favor CIN3 regression. However, the poor prognosis associated with *TDO2* expression in cancer lesions, could indicate that immune evading mechanisms may have outplayed the anti-tumorigenic role of TDO.

CD38 is a type II transmembrane glycoprotein found on the surface of many immune cells including macrophages, CD4^+^, CD8^+^, B lymphocytes, and natural killer cells and is known to activate B and T cells. In addition, CD38 has enzymatic activity as a NADase [47] and regulates diverse pathways such as signal transduction, cell adhesion, cyclic ADP-ribose synthesis, and cell differentiation and activation. In cervical cancer, CD38 has been linked to increased proliferation, inhibition of apoptosis [48] and PI3K/AKT/mTOR signaling activation [49]. Yet, we found that *CD38* associates with CIN3 regression. In line with our findings, *CD38* associates with favorable prognosis in the TCGA cervical cancer cohort (*p* = 0.008, https://www.proteinatlas.org/, accessed on 11 August 2021). One explanation for this discrepancy could be that even though CD38 has oncogenic properties, elevated CD38 levels is also a marker for infiltrating macrophages, natural killer cells, and T and B cells, which in general is favorable for prognosis. Indeed, in hepatocellular carcinoma, CD38^+^ macrophage density associates with improved prognosis [50] and sensitivity to PD-1/PD-L1 immunotherapy [51]. Thus, testing its performance in predicting immunotherapy responses and assessing the prognostic impact by methods such as IHC, is warranted for cervical cancer.

CCL3 and CCL5 are chemokines involved in recruitment of a variety of leukocytes to inflammatory sites, including CD8^+^ T cells, macrophages, eosinophils, and basophils. These chemokines have both anti- and pro-cancer properties [52]. In a study investigating 23 cervical carcinomas, CCL5 levels were found to increase with advancing disease state [53]. However, *CCL5* mRNA levels associate with good prognosis in the cervical cancer TCGA dataset (*p* = 0.002, https://www.proteinatlas.org/ accessed on 11 August 2021) in line with our findings. In cervical precursor lesions, CCL3 expression has been found to increase with increasing premalignant grade. It is presumable that these two chemokines, like TDO and CD38, are markers for adaptive immune responses and that their expression increases with advancing CIN stages aiding as anticancer cytokines. However, the role of CCL3 and CCL5 in cervical cancer needs to be further explored.

Perforin-1 is encoded by the *PRF1* gene and is a pore forming cytolytic protein found in granules of cytotoxic T lymphocytes and natural killer cells. Perforin is known as a key enabler of granzyme-induced apoptosis of target cancer cells [54]. In cervical cancer, down-regulation of perforin has been linked to immune escape [55]. In CIN lesions, decreased perforin granule release in CD8^+^ T cells resulted in more severe lesions [56]. These studies are all in line with our findings that *PRF1* is upregulated in CIN3 regression. Whether perforin expression could be used solely as an IHC detectable marker for CIN2-3 regression warrants further investigation.

GSEA analyses revealed that the Hallmark gene sets “E2F targets” and “G2M checkpoint genes” were significantly enriched in persistent CIN3. While the “G2M checkpoint genes” defines genes associated with G2/M progression, the “E2F targets” include gene linked to G1/S transition. In a non-proliferative state, E2F is bound to the tumor suppressor Rb which inhibits E2F signaling. When the HPV oncoprotein E7 binds to Rb, E2F becomes activated which in turn enables G1/S transition [57]. Within the C2 curated gene sets, 10 out of the top 20 genes sets were related to proliferation of cancer. Microtubules are important for proliferation of cells, and within the GO collection, gene sets related to microtubules were enriched in persistent CIN3. Hence, the GSEA results are unambiguously indicating that proliferation is a central characteristic of persistent CIN3 and that HPV oncoproteins, such as E7, are important drivers for increased proliferation. When E2F is released from Rb, protein levels of the tumor suppressor p16 increases due to a positive feedback loop. This can be detected on IHC and p16 positivity is regarded as a diagnostic marker for HPV infection [58,59]. Interestingly, we found that low regression signature score (associating with persistent CIN3) linked to p16 positivity. This could thus indicate a stronger HPV signaling in the persistent CIN3 cells.

When we applied the regression signature on the cancer cohort, we found that a high score associated significantly to good survival, small tumors, and no vascular space invasion. Vascular space invasion is a prognostic marker of poor survival [31] and lymph node metastases [60] in cervical cancer. Large tumors are also strongly linked to poor prognosis. These results confirm the theory that the presumably less aggressive CIN3 regression lesions share common features with less aggressive cervical carcinomas, whereas persistent CIN3 lesions resembles cancer lesions associated with poor survival.

By GSEA, we discovered that the tumors with high regression signature score and good prognosis were significantly linked to immune activation, immune cell infiltration and increased ESTIMATE immune score. Moreover, these tumors also expressed higher levels of the major histocompatibility complex Class II receptor HLA-DQB1 known as a marker for immune activation, infiltrating immune cells and good prognosis [30]. In order to progress, cancers and precancerous lesions need to bypass immune control. Reduced HLA Class I and II expression in HPV infected tissue may be linked to the HPV oncoprotein E5, which prevents presentation of MHC molecules instigating an immune evading mechanism [25]. In the less aggressive cancer lesions, HLA molecules may still actively express antigens allowing CD8^+^ T cells to identify the pathological cells and thereby restraining the tumor from uncontrolled growth. This may indicate that tumors that resemble the CIN3 regression lesions are, in addition to being less aggressive, also associated with immune activation and immune cell infiltration. When regression from CIN3 occurs, the immune cells overcome the immune evading and proliferative mechanisms initiated by the HPV infection. Our results indicate that these mechanisms are also in play in less aggressive cervical carcinomas decelerating further invasion or metastatic spread.

A limitation of the study is the relatively short observational period between biopsy and cone excision as regression can take from months to years [6,61,62]. However, a longer observational period in the CIN2-3 patients would have been unethical, considering the risk of progression to cancer. Furthermore, the sample size was moderate, which limits the separate interpretation of prognosis in some of the subgroups (i.e., smoking, CIN grade 2 or 3, CIN or cervical cancer in 1st or 2nd degree relatives). An increased sample size would increase the statistical power to identify differentially expressed genes. We were not able to identify differentially expressed genes that remained significant after multiple testing. Still, when applied as a signature, the six genes displayed high predictive power. In addition, Oncomine only includes a subset of 398 immune response related genes, thus other transcriptomic differences between CIN3 regression and persistent CIN3 will go undetected. Another challenge in the interpretation of gene expression data is the lack of spatial and cellular differentiation. To capture the complexity of immune regulation, differentiation between the epithelial, immune related and stromal cells, and the structural composition of the lesions and surrounding tissue, need to be characterized. In the cancer cohort, the application of the gene signature was suboptimal as one gene within the six-gene signature was not included in the transcriptomic cancer cohort data. Still, as a group, the five remaining genes proved useful in predicting disease-specific survival.

Strengths of the current study include the prospective sample collection and close follow-up. Stavanger University Hospital is the only hospital in the region recruiting patients to this study, ensuring a uniform inclusion strategy, and minimizing selection biases. The cervical dysplasia cohort consisted of healthy women with a first-time onset histologically (p16/Ki67 supported) confirmed CIN3, evaluated by two independent pathologists. All patients were followed up after cone excision, and none of the women diagnosed with regression experienced recurrence with high-grade CIN (CIN2/3) during follow-up. Furthermore, although the precancer cohort size is small and the transcriptomic data are confined to 398 immune response related genes, the unprecedented nature of this study and the acquisition of high-value lesions with confirmed regression, makes it highly relevant as a starting point for development of prognostic tools for CIN3 regression. The cervical cancer samples applied for transcriptomic data analysis were collected prospectively in a population-based setting. Extensive clinicopathological characterization and follow-up enabled validation and exploration of results obtained in a precancerous setting on a cancer cohort. By this approach, we have identified previously undetected molecular similarities as well as differences between CIN3 regression, persistent CIN3 and cervical cancer. We consider this study as an important first step towards the development of a clinically applicable pre-operative test for identifying CIN3 regression and hence avoid overtreatment. However, these findings should be validated in larger cohorts.

The development of CIN and cervical cancer is driven by a complex interaction between host and viral factors. Clinicopathological factors including HPV genotype, tobacco smoking, hormonal contraceptives, socioeconomic aspects, reproductive, and sexual factors have been reported as significant for CIN development [63,64,65,66]. Yet, only a few studies have investigated potential prognostic markers for CIN regression, and predictive power have been identified for HPV genotype [67], microbiota [68], age [69], parity, and smoking status [70]. CIN2 is known to have a higher probability of regression, and whether conservative treatment should be recommended for these patients is controversial. In Denmark, national guidelines have recently been changed allowing more than half of women diagnosed with CIN2 conservative treatment. Interestingly, half of these women experienced lesion regression [71], underlining the fact that conservative treatment could be an adequate approach for many CIN2 and potentially CIN3 patients. We present a regression signature that could contribute to risk stratification. However, while our regression signature was identified in CIN3 patients, all the above-mentioned studies were performed on CIN2 lesions. Hence, this study generates novel insights into possible models predicting CIN3 regression. Furthermore, the presented results highlight mechanisms driving persistent CIN3 lesions and provide clues to develop new treatment strategies for patients with CIN2-3.

## 5. Conclusions

CIN2-3 is characterized by a high spontaneous regression rate. Yet, currently, histological evaluation is inadequate to distinguish CIN2-3 lesions that will regress from those that will persist or progress [72]. To our knowledge, this is the first study to date, investigating broad immunological differences between CIN3 regression and persistence. By performing the Oncomine assay, we propose a six-gene signature from 398 immune related genes, predicting regression with a sensitivity of 91% (AUC = 0.85). Furthermore, LCK, Perforin, and CD38 were identified as potential prognostic IHC biomarkers and should be evaluated in larger patient cohorts of both precancer and cancer patients. By unravelling the immune profile of CIN3 regression versus persistence and by comparing this to a large cancer cohort, we have contributed to the etiological understanding of cervical carcinogenesis and have unraveled potential vulnerabilities that may be exploited in future conservative treatment strategies for CIN2-3. Some immune modulating treatments including local use of probiotics, interferon, and biological agents such as Imiquimod and 5-Fluorouracil, have already proven clinical benefit for CIN2-3 regression [73,74,75]. This focus on enabling CIN regression as a possible endpoint instead of quick surgical intervention is a welcoming approach to avoid overtreatment and underpins the growing importance of identifying robust markers for better patient stratification. The potential for development of prognostic markers as well as therapeutics utilizing these differences needs to be further explored in larger and independent high-grade CIN and cervical cancer patient cohorts.

## Data Availability

All data of this study are available within Appendix A or by reasonable request to the corresponding author, if in compliance with the general data protection regulation (GDPR).

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
