# Peer review of "A Gene Signature Identifying CIN3 Regression and Cervical Cancer Survival"

_cancers, 2021, doi:10.3390/cancers13225737_

Round 1

Reviewer 1 Report

This paper describes the identification of common carcinogenic drivers in precursor and cancer lesions and detection of potential biomarkers that can identify CIN3 lesions likely to progress into cancer. The manuscript is excellently written, easy to read and to understand and provides a new insight into a potential problem. Therefore, it deserves to be published in “Cancers” journal. This reviewer has only a few minor suggestions:

  1. Page 1, line 42 – “Low signature score associated…” should be changed to “Low signature score was associated...”
  2. Page 2, line 71 - “results” should be changed to “result”
  3. Page 14, line 519 – “5-Fluorourasil” should be changed to “5-Fluorouracil”

Author Response

Please find your comments and our specific responses below. Please find the attached rebuttal letter for a complete response to the editor and both reviewers. We thank you for your thorough and significant contributions as a reviewer!

Reviewer 1

General comment: This paper describes the identification of common carcinogenic drivers in precursor and cancer lesions and detection of potential biomarkers that can identify CIN3 lesions likely to progress into cancer. The manuscript is excellently written, easy to read and to understand and provides a new insight into a potential problem. Therefore, it deserves to be published in “Cancers” journal.

Response: We thank the reviewer for this feedback.

Specific comments:

Comment 1: Page 1, line 42 – “Low signature score associated…” should be changed to “Low signature score was associated...

Response 1: This has now been corrected in the manuscript.

Comment 2: Page 2, line 71 - “results” should be changed to “result”

Response 2: This has now been corrected in the manuscript.

Comment 3: Page 14, line 519 – “5-Fluorourasil” should be changed to “5-Fluorouracil”

Response 3: This has now been corrected in the manuscript.

Reviewer 2 Report

This paper addresses the clinical management of women that have been identified as carriers of high-risk HPV in screening, and diagnosed with CIN2/3 followed in punch-biopsy. The organized screening based on HPV has resulted in a higher number of women in need of a follow-up than when cytology was used as the primary screening tool. At the same time the nomenclature has changed and CIN2/3 are now classified as HSIL. In most clinics women with HSIL are managed either by conization and histology, or in some cases only monitored b y colposcopy. Since some HSIL will regress, using conization as a standard treatment will result in substantial overtreatment.

This paper focus on the identification of a gene expression signature that could be used to distinguish women with HSIL on a punch biopsy, that are likely to regress spontaneously, from those that will progress to cancer unless treated (with conization).

This is an important topic, and could potentially lead to less over-treatment.

Central to this attempt is the identification of women (and samples) that will either regress and progress from an initial CIN2/3 (or HSIL) diagnosis in the punch. This is also one of the weak points in the paper. Instead of following individual women and the change in gene expression pattern when going from HSIL to LSIL, they define punch-biopsy as the baseline sample and classify the women in two groups based on conization histology. This is a crude approximation and may lead to differences which are more related to the two tissue sampling procedures than true regression. I understand the difficulties involved, but given that punch tissue material would been available for the analysis this would have made for a much more powerful study design. Also, as stated by the authors in the discussion the time between punch biopsy and conization is quite short (for ethics reasons), and there is no data included on the clinical follow-up of these women after conization. Did any of these women come back for follow-up or what happened to them?  

The first gene expression analysis identified 6 genes with modest changes in expression levels. However, the study groups are quite small which results only in a modest statistical power. It was not clear to me if the p-values for the individual genes in the comparison between groups have been corrected for multiple testing? The authors state that “ All probability  values were two-sided and considered statistically significant if <0.05.”  A total of almost 400 genes were studied and correcting for this number (Bonferroni-correction), none of the p-value of the individual genes would remain significant. How can the authors claim that their results are valid?

The authors also examine the “prognostic value” of the score in a different set of patients, but then only five of the six genes are used. This makes a difference in the total score value, and it is impossible to see how the threshold for high and low score are related in using the six- and five-gene models.

Author Response

Please find your comments and our specific responses below. Please find the attached rebuttal letter for a complete response to the editor and both reviewers, also including new analyses. We thank you for your thorough and significant contributions as a reviewer!

Reviewer 2

Comment 1: This paper addresses the clinical management of women that have been identified as carriers of high-risk HPV in screening, and diagnosed with CIN2/3 followed in punch-biopsy. The organized screening based on HPV has resulted in a higher number of women in need of a follow-up than when cytology was used as the primary screening tool. At the same time the nomenclature has changed and CIN2/3 are now classified as HSIL. In most clinics women with HSIL are managed either by conization and histology, or in some cases only monitored by colposcopy. Since some HSIL will regress, using conization as a standard treatment will result in substantial overtreatment.

This paper focus on the identification of a gene expression signature that could be used to distinguish women with HSIL on a punch biopsy, that are likely to regress spontaneously, from those that will progress to cancer unless treated (with conization).

This is an important topic, and could potentially lead to less over-treatment.

Response 1: We thank the reviewer for these comments regarding the relevance of the study.

Comment 2: Central to this attempt is the identification of women (and samples) that will either regress and progress from an initial CIN2/3 (or HSIL) diagnosis in the punch. This is also one of the weak points in the paper. Instead of following individual women and the change in gene expression pattern when going from HSIL to LSIL, they define punch-biopsy as the baseline sample and classify the women in two groups based on conization histology. This is a crude approximation and may lead to differences which are more related to the two tissue sampling procedures than true regression. I understand the difficulties involved, but given that punch tissue material would been available for the analysis this would have made for a much more powerful study design. Also, as stated by the authors in the discussion the time between punch biopsy and conization is quite short (for ethics reasons), and there is no data included on the clinical follow-up of these women after conization. Did any of these women come back for follow-up or what happened to them?

Response 2: We thank the reviewer for addressing the issue regarding choice of samples and for proposing an interesting alternative approach to investigate biological processes of regression. When considering the research question that we had − to find potential clinically applicable pre-operative prognostic markers for regression to avoid overtreatment − we believe that we chose the most appropriate approach. A short discussion regarding samples (including the time between punch biopsy and conization) is already included in the paper lines 474-479. If the reviewer believes this section should be extended, we are open to do so.

We agree with the reviewer that data on follow-up after conization should have been included and have included this on all patients and added this as a new supplementary table S1. In addition, the following text has been included in the methods (Section 2.1, Line 131-134): “All patients were extensively followed up after cone excision. Median follow-up time after cone excision was 1886 days (range 119-5173 days) (Supplementary Table S1). No patients with confirmed regression in the cone excision specimen experienced recurrence with high grade CIN (CIN2/3) during follow-up.”

Comment 3: The first gene expression analysis identified 6 genes with modest changes in expression levels. However, the study groups are quite small which results only in a modest statistical power. It was not clear to me if the p-values for the individual genes in the comparison between groups have been corrected for multiple testing? The authors state that “ All probability  values were two-sided and considered statistically significant if <0.05.”  A total of almost 400 genes were studied and correcting for this number (Bonferroni-correction), none of the p-value of the individual genes would remain significant. How can the authors claim that their results are valid?

Response 3: We thank the reviewer for addressing this. We have now performed multiple testing on all genes in the signature and included the data in a new supplementary table S2 and described this in the methods (Section 2.8., Line 255-256).

We would like to emphasize that even though not all genes remain significant after adjusting for multiple testing, they possess high predictive power when considered together in a signature. The findings from the cancer cohort where high signature score associates with good survival, reinforces this finding. We have included a section in the paper discussing the potential role of each gene in CIN3 disease fate. Based on a joint conclusion, also considering the p-value from the differential gene expression analysis, we conclude on three genes (LCK, PRF1 and CD38) that would be of particular interest to investigate further.

This is the first study that identifies differential gene expression patterns in regressing versus persistent CIN3 and proposes a gene-expression based clinically applicable prognostic tool for identifying regressing CIN3. We consider our study as an important first step towards the development of a clinically applicable test for identifying CIN3 regression, however we agree that our findings should be validated in larger cohorts preferably with a standardized intervals between punch biopsy and conization. We have extended the limitations section to include these reflections (Line 479-480 & Line 506-509).

Comment 4: The authors also examine the “prognostic value” of the score in a different set of patients, but then only five of the six genes are used. This makes a difference in the total score value, and it is impossible to see how the threshold for high and low score are related in using the six- and five-gene models.

Response 4: We thank the reviewer for this comment. Optimal gene signature cut-off values for prediction of CIN3 regression and cervical cancer prognosis were identified from ROC curves applying the Youden index with regression as outcome in the CIN cohort and disease-free survival as outcome in the cancer cohort. We have included an explanation for this in the methods (Section 2.8., Line 264-267).

One of the genes (CCL3) within the six-gene signature (CIN cohort, Oncomine, containing 398 genes) was not part of the L1000 dataset (Cancer cohort, containing 12328 genes). Hence, a 5-gene signature was created within the cancer cohort. We have reviewed the manuscript and made changes to explain this clearer to the reader (Section 3.3, Line 314-319).

To investigate whether CCL3 had a significant contribution to the main results in the CIN cohort, we have now compared the signature with and without CCL3. This yielded a Pearson Correlation Coefficient of 0.98, p<0.001 (see below). Furthermore, when reduced to five genes, the signature had an area under the ROC curve of 0.86 (p<0.001) with the same sensitivity and specificity to predict regression as the 6-gene signature. We decided to still include CCL3 in the proposed signature because when applying the cut off p<0.05 and Fold Change > |1.75|, all these six genes were within this threshold, and should thus be included. Furthermore, the signature is primarily designed to predict CIN3 regression, and we aimed to identify a signature that was optimal for that exact purpose. The investigation within the cancer cohort is a validation of these findings and an attempt to support the CIN cohort findings and to reveal mechanisms shared by CIN3 and cancer lesions. Raw outputs of these analyses can be found in the bottom of this rebuttal letter.

In the revised version of the manuscript, we have now described the results for the comparison of the 5- and 6-gene signatures (Line 319-324). If the reviewer or editor believes that this should be commented further in the manuscript and/or that the herein presented results from this comparison should be added as supplementary figures, please let us know.

Round 2

Reviewer 2 Report

The authors have addressed most of my review concerns. However, in terms of the multiple hypothesis testing issue, the authors have performed an FDR analysis, which shows that given the number of hypothesis (genes) examined, none of the significances found would hold. This information is not mentioned in the paper and the results are hidden in a Supplementary Table. Thus, the reader is given the impression that the statistical analysis performed has identified six genes with different expression levels. I find this handling of statistical evidence unacceptable. The results of the FDR should be included in the results section and discussed as such.

Round 3

Reviewer 2 Report

I think the authors have made appropriate alterations of the manuscript.